# Development and Evaluation of a Novel Deep-Learning-Based Framework for the Classification of Renal Histopathology Images

**DOI:** 10.3390/bioengineering9090423

**Published:** 2022-08-30

**Authors:** Yasmine Abu Haeyeh, Mohammed Ghazal, Ayman El-Baz, Iman M. Talaat

**Affiliations:** 1College of Engineering, Abu Dhabi University, Abu Dhabi 59911, United Arab Emirates; 2BioImaging Lab, Bioengineering Department, University of Louisville, Louisville, KY 40292, USA; 3Clinical Sciences Department, College of Medicine, University of Sharjah, Sharjah 27272, United Arab Emirates

**Keywords:** convolutional neural network, renal cell carcinoma, weakly supervised learning, histopathology

## Abstract

Kidney cancer has several types, with renal cell carcinoma (RCC) being the most prevalent and severe type, accounting for more than 85% of adult patients. The manual analysis of whole slide images (WSI) of renal tissues is the primary tool for RCC diagnosis and prognosis. However, the manual identification of RCC is time-consuming and prone to inter-subject variability. In this paper, we aim to distinguish between benign tissue and malignant RCC tumors and identify the tumor subtypes to support medical therapy management. We propose a novel multiscale weakly-supervised deep learning approach for RCC subtyping. Our system starts by applying the RGB-histogram specification stain normalization on the whole slide images to eliminate the effect of the color variations on the system performance. Then, we follow the multiple instance learning approach by dividing the input data into multiple overlapping patches to maintain the tissue connectivity. Finally, we train three multiscale convolutional neural networks (CNNs) and apply decision fusion to their predicted results to obtain the final classification decision. Our dataset comprises four classes of renal tissues: non-RCC renal parenchyma, non-RCC fat tissues, clear cell RCC (ccRCC), and clear cell papillary RCC (ccpRCC). The developed system demonstrates a high classification accuracy and sensitivity on the RCC biopsy samples at the slide level. Following a leave-one-subject-out cross-validation approach, the developed RCC subtype classification system achieves an overall classification accuracy of 93.0% ± 4.9%, a sensitivity of 91.3% ± 10.7%, and a high classification specificity of 95.6% ± 5.2%, in distinguishing ccRCC from ccpRCC or non-RCC tissues. Furthermore, our method outperformed the state-of-the-art Resnet-50 model.

## 1. Introduction

Kidney cancer is the 9th most commonly occurring cancer in men, and it is less common in women (14th) [1]. Renal cell carcinoma (RCC) is the most common and aggressive type of kidney cancer in adults. It affects nearly 300,000 individuals worldwide annually, and it is responsible for more than 100,000 deaths each year [2]. It develops in the lining of the proximal kidney tubules, where cancerous cells grow over time into a mass and might spread to another organ. The symptoms of RCC are usually hidden and not easily diagnosed.

Many challenges hinder the classification of RCC subtypes, such as the lack of large datasets with precisely localized annotations. Additionally, there is a severe data imbalance since the clear cell subtype comprises the majority of the clinical cases, and the coherency of RCC cells in these different subtypes and the variations in the appearance of the same subtype cells on different resolution levels are also challenges. Recent RCC classification frameworks depend on the precious annotation of pathology digital slides. We propose a weakly supervised learning approach to classify renal cells using image-level labels where no additional costly localized annotations are required. Histopathology images can exhibit dramatically diverse morphology, scale, texture, and color distributions, impeding the extract of a general pattern for tumor detection and subtype classification. The main idea of weakly supervised learning is to translate image-level annotations into pixel/patch-level information. This approach will eliminate the annotations burden on pathologists [3].

The histological classification of RCC is essential for cancer prognosis and the management of treatment plans. Manual tumor subtype classification from renal histopathologic slides is time-consuming and subjective to pathologists’ experience. Each RCC subtype might lead to a different treatment plan, a different survival rate, and a completely different prognoses. We propose an automated CNN-based RCC subtype classification system to assist pathologists with renal cancer diagnosis and prognosis.

### 1.1. Background

RCC is a type of kidney cancer that arises from the renal parenchyma, as shown in Figure 1. Renal parenchyma is the functional part of the kidney, where the tubes filter the blood in the kidneys.

RCC has many different subtypes shown in Figure 2, each with different histology, distinctive genetic and molecular alterations, different clinical courses, and different responses to therapy [2]. The three main histological types of RCC are [4].

**Clear cell RCC (70–90%):** This type of lesion contains clear cells due to their lipid- and glycogen-rich cytoplasmic content. This tumor also depicts cells with eosinophil granular cytoplasm. The imaging of clear cell RCC (ccRCC) reveals hypervascularized and heterogeneous lesions because of the presence of necrosis, hemorrhage, cysts, and multiple calcifications [5]. It has the worse prognosis among other RCC subtypes; its 5-year survival rate is 50–69%. When it expands in the kidney and spreads to other parts of the human body, treatment becomes challenging, and the 5-year survival rate falls to about 10% as per the National Cancer Institute (NCI).

**Papillary RCC (14–17%):** This type occurs sporadically or as a familial condition. Pathologists usually partition this type into two subtypes based on the lesion’s histological appearance and biological behavior. The two subtypes are Papillary type 1 and Papillary type 2, where they have entirely different prognostic factors, where type 2 has a poorer prognosis than type 1. Papillary cells appear in a spindle shape pattern with some regions of internal hemorrhage and cystic alterations [5].

**Chromophobe RCC (4–8%):** Chromophobe is more frequent after the age of 60 and less aggressive than ccRCC. It exhibits the best prognosis among all three RCC subtypes. Under the microscope, it appears as a mass formed of large pale cells with reticulated cytoplasm and perinuclear halos. If sarcomatoid transformation occurs, the lesion develops to be more aggressive with a worse survival rate [5].

Fifteen years ago, pathologists recognized clear cell papillary renal cell carcinoma shown in Figure 3 as the fourth most prevalent type of renal cell carcinoma. It possesses distinct morphologic and immunohistochemical characteristics and an indolent clinical course. When seen under a microscope, it may resemble other RCCs with clear cell characteristics, such as ccRCC, translocation RCC, and papillary RCC with clear cell alterations. A high index of suspicion is needed to accurately diagnose ccpRCC in the differential diagnosis of RCCs with clear cell and papillary architecture. Clinical behavior is highly favorable with rare, questionable reports of aggressive behavior [6].

Recent studies demonstrated that multiple imaging modalities such as magnetic resonance imaging (MRI) and computerized tomography (CT) scans could differentiate ccRCC from other histological types [5]. A new RC-CAD (computer-assisted diagnosis) system presents the ability to differentiate between benign and malignant renal tumors [7]. The system relies on contrast-enhanced computed tomography (CE-CT) images to classify the tumor subtype as ccRCC or not, while the proposed system achieves high accuracy on the given data, the CT imaging method lacks accuracy when the renal mass is small, leading to higher false-positive diagnostic decisions [8,9]. Since pre-surgical CT provides excellent anatomical features with the advantage of three-dimensional reconstruction of the renal tumor, in [10] the authors propose another radiomic machine learning model for ccRCC grading. The presented machine learning model differentiates the low-grade from the high-grade ccRCC with an accuracy of up to 91%. All of the clinical studies conclude that biopsy is the only gold standard for a definite diagnosis of renal cancer. Microscopic images of hematoxylin and eosin (H&E)-stained slides of kidney tissue biopsies are the pathologists’ main tools for tumor detection and classification. Normal renal tissues present round nuclei with uniform distributed chromatin, while renal tumors exhibit large heterogeneous nuclei. Hence, many studies are based on nuclei segmentation and classification for cancer diagnosis.

Deep learning algorithms achieve human-level performance in medical fields on several tasks, including cancer identification, classification, and segmentation. RCC subtype microscopic images exhibit different morphological patterns and nuclear features. Hence, we apply a deep learning approach for RCC detection and subtype classification to extract these essential features.

### 1.2. Related Work

Computational pathology is a discipline of pathology that entails extracting information from digital pathology slides and their accompanying metadata, usually utilizing artificial intelligence approaches such as deep learning. Whole slide imaging (WSI) technology is essential to assess specimens’ histological features. WSIs are easy to access, store, and share, where pathologists can effortlessly embody their annotations and apply different image analysis techniques for diagnostic purposes.

Deep learning (DL) methods generated a massive revolution in medical fields such as oncology, radiology, neurology, and cardiology. DL approaches prove superior performance in image-based tasks such as image segmentation and classification [11]. DL provides the most successful tools for segmenting histology objects such as cell nuclei and glands in computational pathology. However, providing annotations for these algorithms is time-consuming and potentially biased. Recent research aims to automate this process by proposing new strategies that could help provide many annotated images needed for deep learning algorithms [12,13].

Researchers recently deployed deep learning in various CAD systems for cancerous cell segmentation, classification, and grading. Deep CNNs were able to extract learned features that replaced hand-crafted features effectively. Most of the classification models in computational pathology can be categorized based on the available data annotations into two main categories:

**Patch-level classification models:** These models use a sliding window approach on the original WSI to extract small annotated patches. Feeding CNNs with high-resolution WSIs is impractical due to the extensive computational burden. Hence, patch-wise models rely on annotated data as cancerous cells region, normal cells, or specific tumor subtype cells performed by expert pathologists. In [14], the authors developed a deep learning algorithm to identify Nasopharyngeal Carcinoma in Nasopharyngeal biopsies. Their supervised patch-based classification model relies on valuable detailed patch-wise annotated data. Pathologists require more than an hour to annotate only portions of a whole slide image, resulting in a high accuracy patch-level learning model. At the same time, it is a time-consuming process and requires experienced pathologists. They employed gradient-weighted class activation mapping to extract the morphological features and visualize the decision-making in a deep neural network. They applied the patch-level algorithm to classify image-level WSIs, which scored an AUC of 0.9848 for slide-level identification. In [15], the authors present a deep learning patch-wise model for the classification of histologic patterns of lung adenocarcinoma. In their research, a patch classifier combined with a sliding window approach was used to identify primary and minor patterns in whole-slide images. The model incorporates ResNet CNN to differentiate between the five histological patterns and normal tissue. They evaluated the proposed model against three pathologists with a robust agreement of 76.7%, indicating that at least two pathologists out of 3 matched their model prediction results.

**WSI-level classification models:** Most deep learning approaches employ slide-level annotations to detect and classify cancer from histopathology WSIs. These systems follow weakly supervised learning techniques to overcome the lack of large-scale datasets with precisely localized annotations challenges. They usually incorporate a two-stage workflow (patch-level CNN and then slide-level algorithm) known as multiple-instance learning (MIL). In [16], the authors developed a method for training neural networks on entire WSIs for lung cancer type classification. They applied the unified memory (UM) mechanism and several GPU memory optimization techniques to train conventional CNNs with substantial image inputs without modification in training pipelines or model architectures. They achieved an AUC of 0.9594 and 0.9414 for adenocarcinoma and squamous cell carcinoma classification, respectively, outperforming the conventional MIL techniques. The papers mentioned above deployed well-known CNN architectures such as Resnet and Inception-V3. In [17], a simplified CNN architecture (PathCNN) was proposed for efficient pan-cancer whole-slide image classification. PathCNN training data combines three datasets of lung cancer, kidney cancer, and breast cancer tissue WSIs. The proposed architecture converged faster with less memory usage than complex architectures such as Inception-V3. Their model was able to classify the TCGA dataset for RCC subtypes TCGA-KIRC (ccRCC), TCGA-KIRP (Papillary RCC), and TCGA-KICH (Chromophobe RCC), with AUC 0.994, 0.994, and 0.992, respectively. However, they used a large dataset of 430 kidney WSIs partially annotated by specialists. In [18], the researchers presented a semi-supervised learning approach to identify RCC regions using minimal-point-based annotation. A hybrid loss strategy utilizes their proposed classification model results for RCC subtyping. The minimal-point-based classification model outperforms whole-slide based models by 12%, although it requires partially annotated data, which is not always accessible and more subjective to human errors.

Attention-based models have been utilized in computational pathology and presented comparable results to conventional multiple-instance learning (MIL) approaches. However, we are proposing a multiscale MIL framework to elevate the classification performance of the conventional MIL methods. Attention mechanisms can be employed to provide a dynamic representation of features by assigning weights to each recognized feature, improving interpretability and visualization [19]. A clustering-constrained attention multiple-instance learning (CLAM) system was proposed in [20]. A weakly-supervised deep-learning approach employs attention-based learning to highlight sub-regions with high diagnostic value for more accurate tumor subtype classification. The system tests and ranks all the patches of a given WSI, assigning an attention score for each patch, which declares its importance to a particular class’s overall slide-level representation. The developers tested CLAM on three different computational pathology problems; renal cell carcinoma (RCC) subtyping is one of them. Applying a 10-fold macro-averaged one-vs.-rest mean test produced an accuracy of 0.991 for the three-class renal cell carcinoma (RCC) subtyping of papillary (PRCC), chromophobe (CRCC), and clear cell renal cell carcinoma (CCRCC). The attention model of CLAM helps to learn subcategory features in the data while simultaneously filtering noisy patches. However, CLAM did not evaluate the effect of the clustering component. Therefore, replacing the single-instance CNN with the MIL design, incorporating clustering in an end-to-end approach is recommended [21]. Furthermore, CLAM disregards the correlation between instances and does not appraise the spatial information between patches [22]. Therefore, in our proposed system, we extract overlapping patches to maintain the connectivity of the renal tissue and avoid any loss of contextual and spatial information. In [23], the authors implemented the CLAM model for prostate cancer grading and compared multiclass MIL with CLAM and their proposed system. The multiclass MIL outperformed CLAM in terms of classification accuracy.

Proteomics data can be incorporated with histopathology images through machine learning in RCC diagnosis [24]. The proposed proteomics-based RF classifier achieved an overall accuracy of 0.98 (10-fold cross-validation results) in distinguishing between ccRCC and normal tissues and an average sensitivity of 0.97 and specificity of 0.99, while the histology-based classifier scored an accuracy of 0.95 on the test dataset. However, the proposed system was not generalized by testing on different datasets, while it highlights the importance of investigating the predictive relationships between histopathology and proteomics-based diagnostic models.

Recent research employs multiscale WSI patches classification approaches to reflect the pathologist’s actual practices in classifying cancerous regions. To provide his precise evaluation, the pathologist needs to examine the slides at different magnification levels to utilize various cellular features at different scales. In [25], the authors proposed a multiscale deep neural network for bladder cancer classification. The model can effectively extract similar tissue areas using the ROI extraction method. Their multiscale model achieved an F1-score of 0.986; however, their patch-based classification model uses carefully annotated data by expert pathologists. On the other hand, the researchers in [26] proposed another multiscale MIL CNN for cancer subtype classification on slide-level annotation, which is similar to our research. Their proposed system addressed malignant lymphoma subtype classification. The multiscale domain-adversarial MIL approach provided an average accuracy of 0.871 achieved by 5-fold cross-validation. Authors in [27] proposed a state-of-the-art semantic segmentation model for histopathology images. HookNet utilizes concentric patches at multiple resolutions fed to a series of convolutional and pooling layers to effectively combine information from context and WSI’s details. The authors present two high-resolution semantic segmentation models for lung and breast cancer. HookNet outperforms single-resolution U-Net and other conventional multi-resolution segmentation models for histopathology images.

In this research work, we propose a multiscale learning approach for renal cell carcinoma subtyping. We are the first study to apply the proposed idea of combining the decisions of three CNNs in order to provide high accuracy in histopathology classification. We address the problem of choosing the optimal patch size for pathology slides classification while imitating the pathologists’ practice. Since the small patch size closer to the cell’s size will embody different features than the larger patches that maintain the connectivity between the different cells. The main contributions of our proposed framework are as follows:i.The final classification decision is obtained by a multiscale learning approach to integrate the global features extracted from the large-size patches and the local ones extracted from the smaller patches.ii.The end-to-end system framework is fully automated; hence, hand-made feature extraction methods are not required.iii.The decision fusion approach guarantees the discarding of the patches that do not represent the RCC subtype features that we are looking for in our diagnosis, which significantly improves the classification accuracy.

## 2. Materials

Our dataset was acquired from the Bioengineering Department at the University of Louisville (Institutional Review Board (IRB) number: 22.0415) and included high-resolution digital WSIs of H&E stained renal tissue slides. The WSIs were annotated on the slide level by experienced pathologists. A total of 52 renal tissue WSIs annotated on slide-level with four classes (25 ccRCC, 15 ccpRCC, 7 renal parenchyma, and 5 fat tissues). Two classes represent two distinct renal tumor types: ccRCC and ccpRCC, while the rest two classes represent non-cancerous tissues, either parenchyma or fat tissues. Samples of the four different classes of digital slides are shown in Figure 4.

## 3. Methods

Our proposed system for renal cancer histologic sub-types classification using a weakly supervised multiscale learning approach is shown in Figure 5.

The proposed system pipeline consists of multiple stages toward the final classification. It is capable of automatically classifying the extracted tissue cells to either normal cells (fat or Parenchyma) or cancerous cells of ccRCC or ccpRCC.

### 3.1. Data Pre-Processing

The biopsy tissue under the microscope appears transparent; therefore, we use a stain such as H&E to depict the tissue content in different colors. The pathologists treat the tissue samples as stains where any variation in stain color might affect their diagnosis [28]. Staining conditions vary greatly depending on the specimen conditions and the hospital that acquired the specimen. Thus, WSI color normalization is crucial in any histopathology image classification or segmentation task. There are multiple methods for the stain normalization of histopathology images to reduce the effect of color variations. We investigated the performance of different available approaches, such as RGB histogram specification (Reinhard et al. [29], Macenko et al. [30], and Khan et al. [31]) to choose the best method for our proposed system. The RGB histogram-based normalization approach is used for our proposed system since it was proved in the literature to provide higher accuracy on kidney datasets with the most negligible computation complexity [28]. The approach starts by converting the RGB image to ℓαβ space and then maps the image histogram to the target image histogram. Finally, we convert the image back to the RGB color space. The target image was chosen from within the same dataset to ensure that both source and target images have similar statistics. The stain normalization process applying RGB histogram specification is shown in Figure 6 to demonstrate the effect of the applied color normalization approach to eliminate the stain variation effect on our CNN training.

### 3.2. Tiling

Processing a large size WSI with deep learning approaches is computationally intractable [32]. One WSI can produce hundreds of thousands of 400 × 400 tiles. Most available algorithms select random tiles or patches from each subject slide. Training the CNN on selected patches will reduce the computational burden of processing all possible extracted tiles. However, applying a sliding window to extract small patches is less efficient since we lose valuable information about the degree of overlap between cells. On the other hand, using a large window increases the number of model parameters and training time.

Our proposed pipeline extracts overlapping patches from each training subject’s WSI. We extract the patches using a sliding window horizontally and vertically by skipping 50 pixels as shown in Figure 7. This approach preserves the contextual information in every patch and the overall visual context instead of selecting random small patches. We discard any tiles with a background of more than 25% since it does not hold enough information to be utilized in the training process. For testing, we extract multiscale concentric tiles of 100 × 100, 200 × 200, and 400 × 400 aligned patches, sharing the same center pixel as shown in Figure 8. The trained, multiscale CNNs will classify each small test patch and its parent larger concentric patches in a pyramidal approach to extract its inherited features. Samples of the four different classes of extracted patches after color normalization are shown in Figure 9.

### 3.3. Training: Multiscale Classification

We utilize the MIL algorithm in our proposed system, which is a weakly supervised learning approach leveraged in histopathology classification [33]. The MIL algorithm starts by arranging the training instances in sets, called bags, and labeling the entire bag. The extracted small patches are the instances in our system, and the bags are the training slides. We propose a multiscale system that mimics the pathologists’ manual classification tasks. They observe the renal tissue on different magnification levels to compare and contrast the different cellular features in order to make a decision.

We train three CNNs on different input patches scales; 400 × 400, 200 × 200, and 100 × 100. The CNN architecture in our pipeline follows the ResNet-50 architecture (see Figure 10), where the identity shortcut connections enable training a deeper network faster while avoiding vanishing gradient problems [34]. Many researchers leveraged the ResNet architecture in histopathology since it employs more complex connections between the residual blocks [35,36,37,38]. The ResNet represents deeper architecture compared to VggNet and AlexNet. However, the residual connections between its layers maintain the perceived learning through the whole training process while training faster by increasing the network capacity. We listed the optimal hyperparameters of training our CNNs in Table 1. We utilized early stopping in python to avoid overtraining our CNN. We configured our network training to function under monitoring the model performance. Therefore, the validation accuracy reached the maximum after ten epochs and did not improve further. Moreover, we employed a learning rate control method in python for reducing the learning rate while monitoring the validation accuracy. We reduce the learning rate by a factor of 0.1 once learning stagnates for the patience of two epochs. We performed a hyper-parameters optimization to choose the optimal patch size and optimizer for training our classification model, as shown in Table 2, where the mini-patch size of 64 produced the highest validation accuracy. The Adam optimizer is computationally efficient with fewer memory requirements, which makes it suitable for problems with large data such as our system [39]. The results of testing three different optimizers are shown in Table 3, where Adam outperformed the stochastic gradient descent with momentum (SGDM) and RMSprop. Finally, the trained classifiers will be tested on the test patches to decide the predicted class of these patches.

### 3.4. Testing: Fusion and Features Aggregation

The trained, multiscale classifiers will generate three predicted classes for every single patch in the test dataset (see Figure 11). First, we implement decision fusion following Algorithm 1 to classify each patch as a class-based patch or discarded. Then, the percentage of the overall class-based patches will be mapped per slide to decide the slide-level predicted class, using majority voting. This approach will enhance the system’s classification performance and the ability to adapt to new data.
**Algorithm 1:** Decision Fusion1:**for** Every patch in the test slide **do**2:    Classify its 400 × 400 parent patch using the trained network CNN1 and output YPred13:    Classify its 200 × 200 parent patch using the trained network CNN2 and output YPred24:    Classify that 100 × 100 patch using the trained network CNN3 and output YPred35:    Vote = 06:    Count = 07:    Similarity = 08:    **for** Each Decision in [YPred1, YPred2, YPred3] **do**9:        **if** Count = 0 **then**10:           Vote = Decision11:        **end if**12:        **if** Vote = Decision **then**13:            Count = Count + 114:        **else**15:           Count = Count − 116:        **end if**17:    **end for**18:    **if** Vote ≠ Decision(1) and Vote ≠ Decision(2) **then**19:        Vote = Patch is discarded20:    **end if**21:**return** Vote22:**end for**

## 4. Evaluation Metrics

Researchers use multiple metrics to assess the performance of deep learning models tailored for medical image analysis. The evaluation process primarily starts by obtaining the confusion matrix to visualize the performance of the algorithm and to calculate the different evaluation metrics [40]. The confusion matrix’s rows represent the number of test samples of actual results, and the columns represent the test samples of predicted results of the algorithm shown in Figure 12. The ratio of correctly classified samples to the total number of samples in the test data is the accuracy. This metric is one of the most widely employed in medical applications of machine learning. However, it is also known to be deceptive in the case of varying proportions between classes. Sensitivity or recall is the ratio between successfully classified positive samples and all samples allocated to the positive class to compute the rate of positive samples correctly classified. This metric is one of the most significant in medical data analysis. It reflects how likely a test can accurately detect a tumor/disease when it is present in a patient resulting in a high recall. The specificity is the negative class version of the sensitivity, indicating the percentage of correctly classified negative samples. It is calculated as the proportion of correctly classified negative samples to all negative samples. It is also crucial in medical investigations since it represents the ability of a test to rule out the presence of a tumor/disease in someone who does not have it [41]. To evaluate our proposed framework, we report the accuracy, sensitivity, and specificity on the patch as well as the WSI level.

## 5. Experimental Results

Multiscale classification and decision fusion approaches are integrated into our proposed system to generalize the classification model and classify new data accurately. To compare the performance of the proposed framework with the single multiscale classifiers (CNN1, CNN2, and CNN3), we applied a 5-fold cross-validation and reported the results in Table 4. The proposed decision fusion approach achieves higher accuracy compared to each of the single classifiers. Many research works utilized an ensemble of CNNs to classify or segment histopathology images such as [42,43]. Hence to highlight the merit of our proposed multiscale ensemble of classifiers, we experimented by combining the decision of three different CNN architectures into a single-scale ensemble of classifiers. To compare our proposed system results, we applied the same 5-fold cross-validation. We reported the results of each single-scale CNN accuracy and the final decision utilizing the same adopted decision fusion algorithm in Table 5. We chose ResNet architecture again for its superior performance in classifying WSI patches as in [44], where Renet18 and Resnet50 outperformed other CNN architectures in multiclass classification problems. Furthermore, MobileNetV2 was adopted in [42,45,46] to classify breast histopathology images achieving comparable performance to deeper architectures with fewer model parameters. Hence, we trained Renset18, Renset50, and MobileNetV2 on our single-scale extracted patches. Comparing Table 4 and Table 5 confirms the importance of our proposed multiscale approach in extracting the local and global features and combining them to produce a more accurate and robust classification decision.

We implemented Resnet-50 architecture without transfer learning to achieve higher accuracy by extracting deeper features in our proposed system. As the ImageNet dataset varies from our renal cancer tissue slides, complete learning will consume more time while outperforming transfer learning results to accomplish our target of designing a robust and highly accurate classification model for renal cancer diagnosis.

To evaluate the generalization of our final proposed system, we applied the leave-one-subject-out (LOSO) cross-validation approach. We summarized the overall classification performance on the patch-level in terms of average accuracy, sensitivity, and specificity in Table 6. The final classification for the whole slide is concluded based on the percentage of correctly classified patches. The WSI class label will be defined based on the class label with the most significant percentage of patches per slide. We report the overall accuracy on the slide level in Table 7.

## 6. Discussion

Our experimental results demonstrate the high accuracy of the proposed classification system on both the patch and slide levels. Implementing multiple algorithms such as multiscale CNNs and the final decision fusion led to these promising results. Color normalization was an essential step at the beginning of the overall system pipeline; without it, the system could not differentiate between renal tumor tissues and non-tumor tissues due to color variations. Our system accurately distinguished fat tissue slides with a 100% accuracy due to the distinct features of the fat tissues from other renal tissues such as parenchyma and RCC cells. On the other hand, ccRCC and ccpRCC have similar morphological features, leading to a more challenging classification task. However, our proposed system could differentiate between these two types of RCC with an accuracy of 92% to 93.3%.

Table 5 shows a comparison of the classification performance of each single-scale classifier and a single-scale ensemble of classifiers. In that table, Resnet50 outperformed Resnet18 and MobileNetV2 in terms of mean accuracy. On the other hand, our proposed decision fusion provides the best mean accuracy.

Table 6 shows a comparison of the classification performance on the patch level. For our proposed technique, we obtained the best classification metrics from the fat class. In other words, our system was optimal with a 100% classification accuracy for the fat class. Furthermore, we obtained the lowest performance from class ccRCC with a 89% classification accuracy. For Resnet50, we obtained the best classification accuracy (96.8%) in the parenchyma class and the lowest classification accuracy (82.1%), again for ccRCC class.

Table 7 shows a comparison of the classification performance on the slide level. Our proposed technique provides the best classification accuracy in the case of the fat class. On the other hand, the lowest accuracy has resulted from parenchyma class. For Resnet-50, the best performance was obtained from ccRCC class, while the lowest performance was obtained from ccpRCC class. Furthermore, it is shown in the table that our proposed system surpassed Resnet-50 in terms of classification accuracy.

Our proposed method uses an ensemble of three CNNs with different input sizes to allow the learning from multiple viewpoints for the underlying slide image. Using an ensemble of three CNNs boosts the classification accuracy and achieves better metrics than using only one CNN such as in the case of existing methods. We compared our proposed method with well-known state-of-the-art networks to provide the reader with a fair judgment of the superiority of our method over other existing techniques that are nowadays used heavily due to their high performance.

The developed system presents an advanced computational pathology technique for WSI analysis. RCC treatment highly depends on distinguishing the tumor sub-type and the grade of the tumor cells. Future development of this work might integrate the proposed classification approach for RCC grading for a complete CAD system that provides the full diagnostic and prognostic details of a given renal tissue.

## 7. Conclusions

This paper proposes an automated deep learning-based classification system for RCC subtyping from histopathology images. Our proposed weakly-supervised approach does not require precise annotations on the patch level, reducing the pathologists’ annotations burden. We developed a multiscale classification framework to extract the local features in the renal cells and the global features in the tissue connectivity. The proposed decision fusion approach outperforms the single CNN classifier’s performance to present a rigorous diagnosis decision. The proposed algorithm addresses the main challenges in computational histopathology using deep learning. The RCC-subtype classification system achieved an overall classification accuracy of 93.0% ± 4.9%, a sensitivity of 91.3% ± 10.7%, and a high classification specificity 95.6% ± 5.2% in distinguishing ccRCC from ccpRCC or non-RCC. Our results confirm the approach’s robustness in identifying and classifying renal cancer subtypes on the WSI level.

## Figures and Tables

**Figure 1 bioengineering-09-00423-f001:**
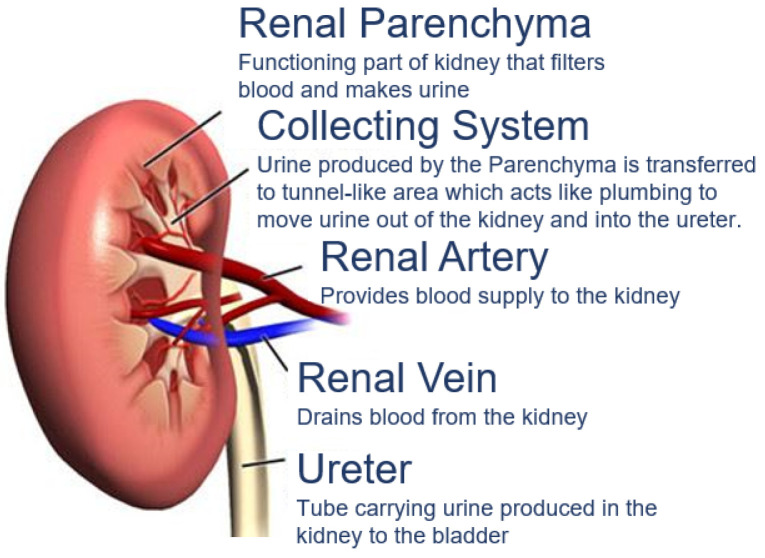
Kidney parenchyma.

**Figure 2 bioengineering-09-00423-f002:**
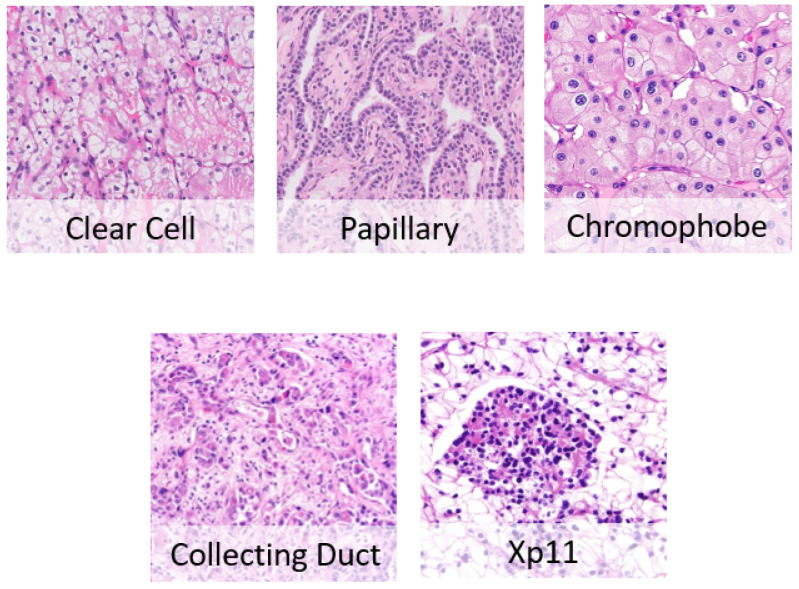
RCC subtypes.

**Figure 3 bioengineering-09-00423-f003:**
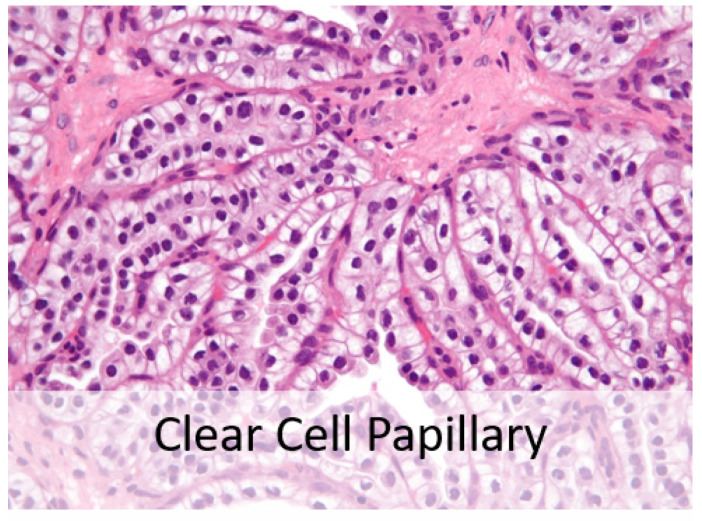
Clear cell papillary RCC subtype.

**Figure 4 bioengineering-09-00423-f004:**
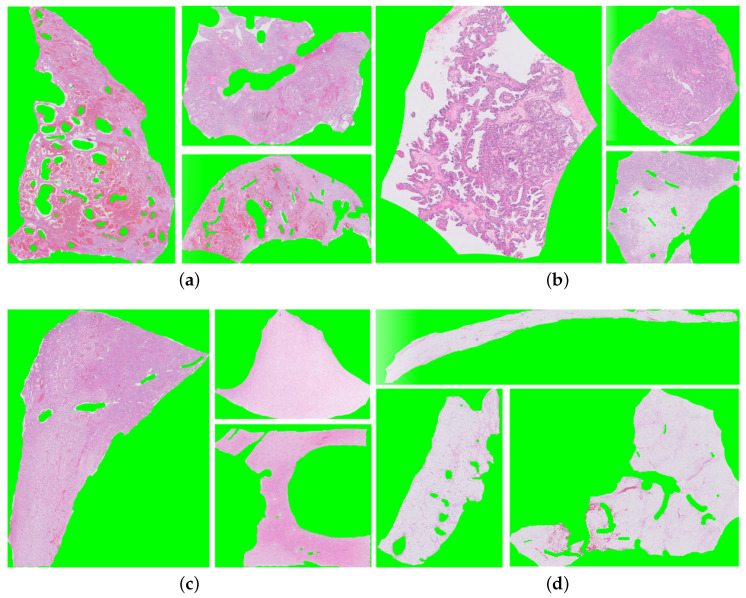
Samples of renal tissue WSIs from different classes (**a**) clear cell RCC WSI; (**b**) clear cell papillary RCC WSI; (**c**) parenchyma WSI; and (**d**) fat WSI.

**Figure 5 bioengineering-09-00423-f005:**
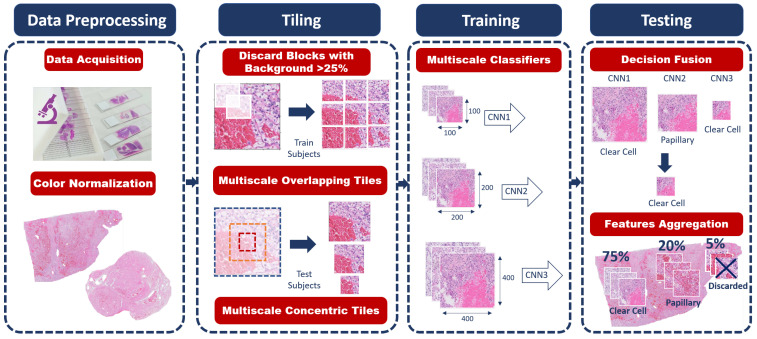
The proposed system pipeline. The first stage is data preprocessing, including color normalization. The second stage is tiling to divide the WSI into small patches. The third stage is training the multiple CNNs. Finally, the last stage is testing to predict the final classification decision.

**Figure 6 bioengineering-09-00423-f006:**
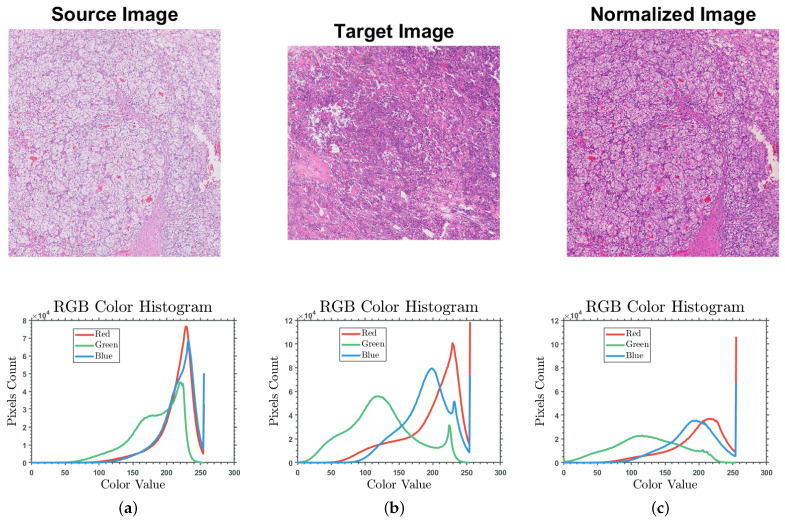
The stain normalization of a clear cell RCC sample (**a**) before color normalization; (**b**) the target sample; and (**c**) after color normalization.

**Figure 7 bioengineering-09-00423-f007:**
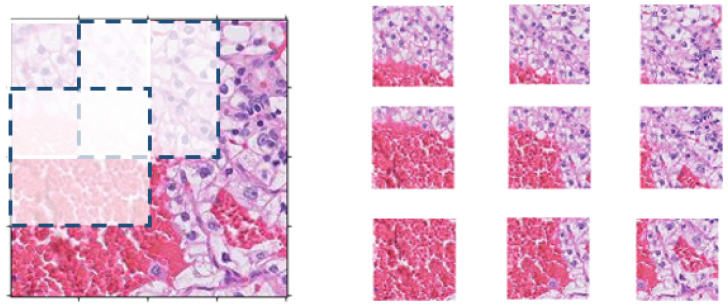
Tiling, extracting overlapping patches of training subjects’ WSIs.

**Figure 8 bioengineering-09-00423-f008:**
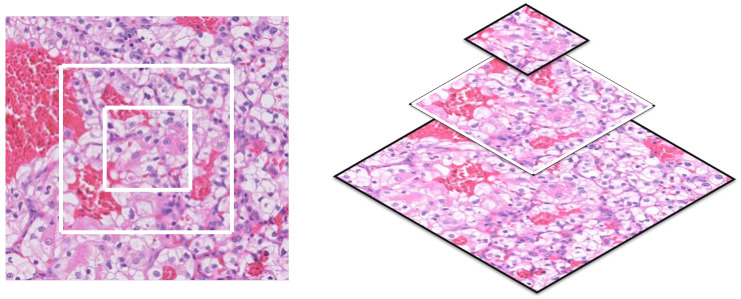
Tiling, extracting concentric patches of test subjects’ WSIs.

**Figure 9 bioengineering-09-00423-f009:**
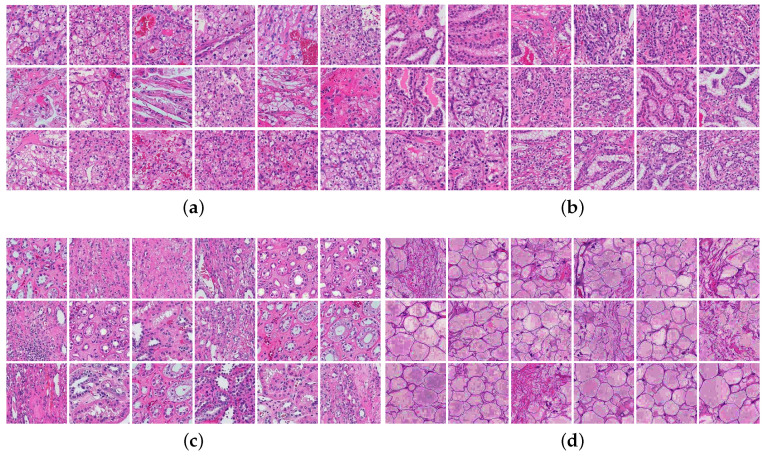
Samples of color-normalized extracted patches from different classes: (**a**) clear cell patches; (**b**) clear cell papillary patches; (**c**) parenchyma patches; and (**d**) fat patches.

**Figure 10 bioengineering-09-00423-f010:**
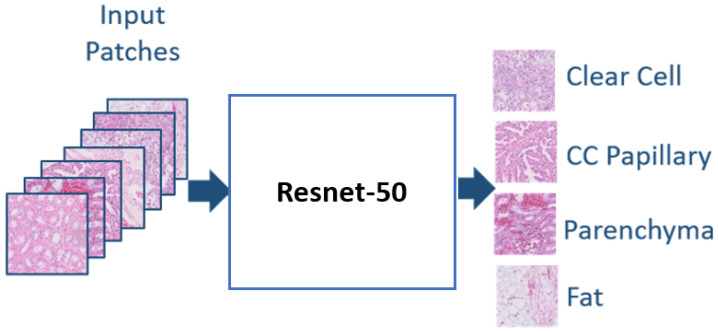
Using ResNet-50 model in the proposed technique.

**Figure 11 bioengineering-09-00423-f011:**
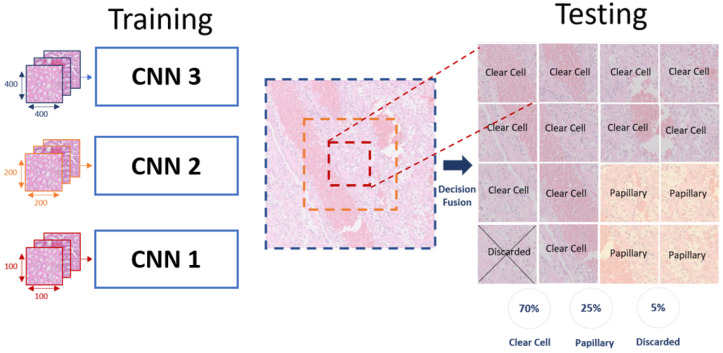
Demonstration of the decision fusion implementation.

**Figure 12 bioengineering-09-00423-f012:**
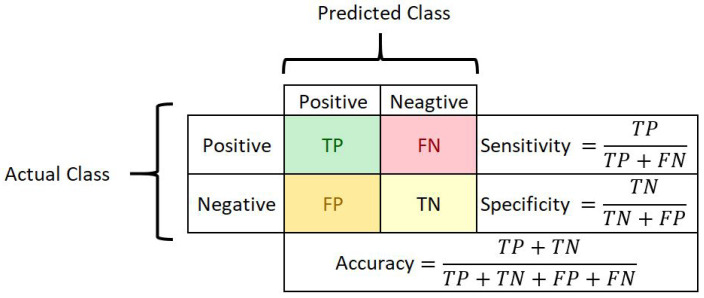
Confusion matrix.

**Table 1 bioengineering-09-00423-t001:** CNN’s optimal hyperparameters.

Hyperparameter	Value
Patch size	64
Number of epochs	10
Learning rate	0.001
Optimizer	Adam
Loss Function	Categorical Crossentropy

**Table 2 bioengineering-09-00423-t002:** CNN’s performance comparison for different patch sizes.

Optimizer	Patch Size	Validation Accuracy %	Learning Rate
Adam	32	84.01	0.001
Adam	64	95.63	0.001
Adam	128	91.68	0.001

**Table 3 bioengineering-09-00423-t003:** CNN’s performance comparison for different optimizers.

Optimizer	Patch Size	Validation Accuracy %	Learning Rate
SGDM	64	90.41	0.001
RMSprop	64	92.00	0.001
Adam	64	95.63	0.001

**Table 4 bioengineering-09-00423-t004:** Comparison of the classification performance of each single-scale classifier and the proposed multiscale ensemble of classifiers.

Classification Performance of the Multiscale Classifiers on the Patch-Level				
**Fold**	**CNN1**	**CNN2**	**CNN3**	**Decision Fusion**
	**Accuracy %**	**Accuracy %**	**Accuracy %**	**Accuracy %**
1	89.10	90.54	95.05	99.77
2	73.33	80.45	76.60	97.42
3	67.28	68.52	66.05	87.42
4	54.20	73.07	67.61	95.52
5	58.18	69.76	55.68	88.66
**Mean**	**68.42**	**76.47**	**72.20**	**93.76**

**Table 5 bioengineering-09-00423-t005:** Comparison of the classification performance of each single-scale classifier and a single-scale ensemble of classifiers.

Classification Performance of the Single-Scale Classifiers on the Patch-Level				
**Fold**	**Resnet18**	**Resnet50**	**MobileNetV2**	**Decision Fusion**
	**Accuracy %**	**Accuracy %**	**Accuracy %**	**Accuracy %**
1	88.29	90.54	87.84	96.59
2	81.73	80.45	82.05	93.67
3	51.85	68.52	74.07	76.85
4	71.02	73.07	66.48	92.29
5	56.36	69.76	54.55	78.35
**Mean**	**69.85**	**76.47**	**73.00**	**87.55**

**Table 6 bioengineering-09-00423-t006:** Comparison of the classification performance on the patch level.

Classification Performance Patch-Level				
**Method**	**Class**	**Accuracy %**	**Sensitivity %**	**Specificity %**
Proposed	Clear Cell	89.0	87.7	93.1
	CC Papillary	92.8	99.8	89.5
	Parenchyma	90.3	77.7	99.9
	Fat	100.0	100.0	100.0
	Mean ± SD	93.0 ± 4.9	91.3 ± 10.7	95.6 ± 5.2
Resnet-50	Clear Cell	82.1	67.4	92.4
	CC Papillary	82.6	89.7	75.9
	Parenchyma	96.8	60.0	98.0
	Fat	96.2	80.0	97.4
	Mean ± SD	89.4 ± 8.2	74.3 ± 13.2	90.9 ± 10.3

**Table 7 bioengineering-09-00423-t007:** Comparison of the classification performance on the slide level.

Classification Performance WSI-Level		
**Method**	**Class**	**Accuracy %**
Proposed	Clear Cell	92.0
	CC Papillary	93.3
	Parenchyma	85.7
	Fat	100
	Mean ± SD	92.8 ± 5.9
Resnet-50	Clear Cell	84.0
	CC Papillary	66.7
	Parenchyma	71.4
	Fat	80
	Mean ± SD	75.5 ± 7.9

## Data Availability

Data could be made available after acceptance upon a reasonable request to the corresponding author.

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
