# Peer review of "Development and Evaluation of a Novel Deep-Learning-Based Framework for the Classification of Renal Histopathology Images"

_bioengineering, 2022, doi:10.3390/bioengineering9090423_

Round 1

Reviewer 1 Report

This paper proposes a novel multiscale weakly-supervised deep learning  approach for RCC subtyping as shown in Fig 5.

The review comments are as follows:

- On the structure of your manuscript, there is not matched with the procedure of Fig. 5.  

- What is the motivation and originality of Data preprocessing, Tiling, Training, and testing as shown in Fig. 5?

- In the experimental results, you used CNN1, Resnet18, Resnet50, and MobileNetV2 in the existing methods. What is the choosing reason in detail? 

Author Response

  • On the structure of your manuscript, there is not matched with the procedure of Fig. 5.

Thank you for your comment. We organized the structure of the methods to match the organization of Fig. 5. Please refer to Page 10 (line 312) and Page 11 (line 345).

  • What is the motivation and originality of Data preprocessing, Tiling, Training, and testing as shown in Fig. 5?

Thank you for your comment. RGB Histogram-based normalization approach is used for our proposed system since it was proved in the literature to provide higher accuracy on kidney datasets with the most negligible computation complexity [28]. Our proposed pipeline extracts overlapping patches from each training subject’s WSI. This approach preserves the contextual information in every patch and the overall visual context instead of selecting random small patches. For training and testing, we are the first study to apply the proposed idea of combining the decisions of three CNNs in order to provide high accuracy in histopathology classification. We have clarified this point in the revised manuscript. Please refer to Section 3.1, Pages 7 and 8, lines 280-283, Section 3.2, Page 8, lines 300-303, and Page 6, lines 238-240.

[28] Roy, S.; kumar Jain, A.; Lal, S.; Kini, J. A study about color normalization methods for histopathology images. Micron 2018 114, 42–61.

In the experimental results, you used CNN1, Resnet18, Resnet50, and MobileNetV2 in the existing methods. What is the choosing reason in detail? 

Thank you for your comment. We chose these networks because of their popularity and efficiency in medical applications. More specifically, we chose ResNet architecture for its superior performance in classifying WSI patches as in [44], where Renet18 and Resnet50 outperformed other CNN architectures in multiclass classification problems. Furthermore, MobileNetV2 was adopted in [42,45,46] to classify breast histopathology images achieving comparable performance to deeper architectures with fewer model parameters. Hence, we trained Renset18, Renset50, and MobileNetV2391 on our single-scale extracted patches and use them for comparison. Please refer to Section 5, Pages 13 and 14, lines 386-392.

[42] Kassani, S.H.; Kassani, P.H.; Wesolowski, M.J.; Schneider, K.A.; Deters, R. Classification of histopathological biopsy images using ensemble of deep learning networks. arXiv preprint arXiv:1909.11870 2019.

[44] Tsai, M.J.; Tao, Y.H. Deep Learning Techniques for the Classification of Colorectal Cancer Tissue. Electronics 2021, 10, 1662.

[45] Wang, C.; Gong, W.; Cheng, J.; Qian, Y. DBLCNN: Dependency-based lightweight convolutional neural network for multi-classification of breast histopathology images. Biomedical Signal Processing and Control 2022, 73, 103451.

[46] Laxmisagar, H.; Hanumantharaju, M. Design of an Efficient Deep Neural Network for Multi-level Classification of Breast Cancer Histology Images. In Intelligent Computing and Applications; Springer, 2021; pp. 447–459

Reviewer 2 Report

This paper proposed a multiscale weakly-supervised deep learning approach for renal cell carcinoma subtyping classification. The detailed comments are listed as follows: 

1.     In Line 181, the full name of MIL is not given until in Line 184.

2.     The link for the dataset should be given.

3.     In Fig. 10, the Resnet-50 model is not illustrated correctly and the skip connection is missed.

4.     CNNs have been used in patch-level classification. What are advantages of the proposed CNN compared to the existing ones?

5.     How to make the final decision fusion?

Author Response

  1. In Line 181, the full name of MIL is not given until in Line 184.

Thank you for pointing this out. We added the full name of MIL, as first mentioned, in line 183, Page 5.

  1.     The link for the dataset should be given.

Thank you for your comment. The data set is not publicly available. We can not share the data publically due to IRB restrictions (the IRB letter is attached). However, we can share it upon request. Please, refer to the Material section, Page 6, line 255.

  1.     In Fig. 10, the Resnet-50 model is not illustrated correctly and the skip connection is missed.

Thank you for your comment. We have corrected and updated Fig. 10 to avoid copyright issues. Please, refer to Figure 10, Page 11

  1.     CNNs have been used in patch-level classification. What are advantages of the proposed CNN compared to the existing ones?

Thank you for your comment. In our proposed approach, we used three CNNs with different input sizes this allows the learning from multiple viewpoints for the underlying slide image. Furthermore, using an ensemble of three CNNs boosts the classification accuracy and achieves better metrics than using only one CNN such as in the case of existing methods. We have clarified this point in the revised manuscript. Please, refer to Section 6, Page 16, lines 438-444.

  1.     How to make the final decision fusion?

Thank you for your comment. The decision fusion is illustrated in Algorithm 1 and Figure 11. The trained, multiscale classifiers will generate three predicted classes for every single patch in the test dataset as demonstrated in figure 11. First, we implement decision fusion following algorithm 1 to classify each patch as a class-based patch or discarded. Then, the percentage of the overall class-based patches will be mapped per slide to decide the slide-level predicted class, using majority voting. We have clarified this point in the revised manuscript. Please, refer to Section 3.4, Page 11, lines 346-351.

Round 2

Reviewer 1 Report

It is not appropriate the responses based on review comments.  Finally, i can not give you the acceptance to publish the publication. 

Reviewer 2 Report

The manuscript has been improved according to the comments.